# Improving CLIP Counting Accuracy via Parameter-Efficient Fine-Tuning

**Ruisu Zhang**                                                                    *rzhang345@wisc.edu*
*Department of Electrical and Computer Engineering*
*University of Wisconsin-Madison*

**Yicong Chen**                                                                    *ychen2229@wisc.edu*
*Department of Electrical and Computer Engineering*
*University of Wisconsin-Madison*

**Kangwook Lee**                                                                   *kangwook.lee@wisc.edu*
*Department of Electrical and Computer Engineering*
*University of Wisconsin-Madison*

**Reviewed on OpenReview:** *https://openreview.net/forum?id=IZrt6hB2sI*

## Abstract

We focus on addressing the object counting limitations of vision-language models, with a particular emphasis on Contrastive Language-Image Pre-training (CLIP) models. Centered on our hypothesis that counting knowledge can be abstracted into linear vectors within the text embedding space, we develop a parameter-efficient fine-tuning method and several zero-shot methods to improve CLIP's counting accuracy. Through comprehensive experiments, we demonstrate that our learning-based method not only outperforms full-model fine-tuning in counting accuracy but also retains the broad capabilities of pre-trained CLIP models. Our zero-shot text embedding editing techniques are also effective in situations where training data is scarce, and can be extended to improve Stable Diffusion's ability to generate images with precise object counts. We also contribute two specialized datasets to train and evaluate CLIP's counting capabilities. Our code is available at `https://github.com/UW-Madison-Lee-Lab/CLIP_Counting`.

## 1 Introduction

Recent advancement of deep learning techniques has led to significant progress in vision-language models (Alayrac et al., 2022; Chen et al., 2023; Radford et al., 2021; Singh et al., 2022; Liu et al., 2023; Li et al., 2023a). One such breakthrough is the development of Contrastive Language-Image Pre-training (CLIP) (Radford et al., 2021), which is trained on a wide range of internet text-image pairs (Schuhmann et al., 2021). CLIP has demonstrated strong performance in zero-shot learning tasks and serves as a backbone for text-to-image generative models like Stable Diffusion. (Rombach et al., 2021).

Despite its extensive deployment, CLIP exhibits limitations in certain areas (Radford et al., 2021; Liu et al., 2021; Thrush et al., 2022; Paiss et al., 2022), such as counting objects in images (Paiss et al., 2023). Counting is a fundamental skill that requires the integration of visual and linguistic understanding, and it plays a crucial role in numerous practical applications. Existing works have attempted to address the object counting limitations with CLIP models or CLIP-based models (Paiss et al., 2023; Jiang et al., 2023a; Mestha et al., 2024; Binyamin et al., 2024). However, these methods often require extensive training on large datasets.

Our work seeks a deeper understanding of CLIP's object counting capability and introduces a data-efficient and compute-efficient approach to enhancing it. We contribute two specialized datasets for training and evaluating CLIP's counting capabilities.

Our key idea hypothesizes that counting knowledge can be abstracted as vectors in the text embedding space, distinct from non-counting information. Our method seeks to find an object-agnostic counting vector that represents the concept of a count (e.g., "five"), which is not necessarily the text embedding of the count word. Once such a vector representation is identified, it is added to the text embedding of the original caption (e.g., "an image of five dogs") to reinforce the counting signal. This idea of finding counting-specific vectors also provides a parameter-efficient and scalable solution that avoids extensive model training, and can be universally applied in tasks that involve counting any given type of object.

In our paper, we develop a learning-based method and several zero-shot methods to obtain the counting representation. First, we employ a counting-specific contrastive loss (Paiss et al., 2023) to train these vectors. Our experiments reveal that this counting vector training not only outperforms traditional full-model fine-tuning methods in terms of counting accuracy but also avoids the issue of the pre-trained CLIP model losing its broad capabilities when fine-tuning the entire model on new data.

Furthermore, in scenarios where direct training data is scarce, we demonstrate that a counting vector can be formulated following simple rules or can be extracted from objects that CLIP counts more proficiently. To demonstrate the benefits of improved counting accuracy of CLIP, we also test the fidelity on text-to-image models. In particular, we test our zero-shot method on Stable Diffusion models (Rombach et al., 2021) and provide examples to show that it helps Stable Diffusion models generate images with precise object counts as specified in the caption.

Our contributions include: (i) a parameter-efficient training method that enhances CLIP's counting accuracy while preserving its capabilities; (ii) zero-shot text embedding editing techniques effective without training data; (iii) two novel datasets for fine-tuning and evaluating CLIP's counting ability; and (iv) a demonstration of how our zero-shot approach guides text-to-image models like Stable Diffusion to generate images with accurate object counts.

## 2 Related Work

**Vision-language models** Vision-language models (VLMs) have achieved significant success in multimodal tasks by training on massive image-text datasets and excelling in zero-shot or fine-tuned downstream tasks (Alayrac et al., 2022; Chen et al., 2023; Radford et al., 2021; Singh et al., 2022; Liu et al., 2023; Li et al., 2023a). In this work, we will focus on the Contrastive Language-Image Pre-training (CLIP) model trained by OpenAI (Radford et al., 2021). CLIP is trained on 400 million image-caption pairs (Schuhmann et al., 2021), using a contrastive objective where matching text-image pairs are optimized to have a low cosine distance, while mismatched pairs are pushed further apart. CLIP has demonstrated notable success across a range of visual tasks due to its zero-shot capabilities. It also underpins text-to-image alignment in generative models like Stable Diffusion (Rombach et al., 2021).

**Limitations of vision-language models on counting** While VLMs perform impressively across various tasks, they struggle with specific challenges, such as counting objects in images (Radford et al., 2021; Liu et al., 2021; Thrush et al., 2022; Paiss et al., 2022), like counting objects within pictures (Paiss et al., 2023). In fact, the object counting problem has always been one of the important issues in the visual question answering (VQA) field, and several studies have attempted to address it (Jiang et al., 2023b; Xu et al., 2023; Zhang et al., 2018; Nguyen et al., 2021; Acharya et al., 2019).

One reason for CLIP's limitations in object counting may stem from its mini-batch contrastive pre-training process. Typically, these models rarely process images of the same object in varying counts within a single training batch. As a result, CLIP models have minimal exposure to learning nuanced differences in object counts during pre-training. While full-batch training could address this issue, it is often prohibitively costly and impractical.

One potential solution during pre-training is to strategically select mini-batches under certain conditions, allowing them to mimic full-batch optimization (Sreenivasan et al., 2023). Using a tailored training set and loss function can better guide models in learning to count. (Paiss et al., 2023; Mestha et al., 2024). For example, Paiss et al. (2023) fine-tunes pre-trained CLIP models using a counting-specific loss on a

counting-relevant dataset filtered from the LAION-400M dataset (Schuhmann et al., 2021). It also introduces a new image-text counting benchmark, *CountBench*, used to evaluate a model's understanding of object counting, which we also utilized in our experiments. Mestha et al. (2024) make further improvements by redesigning the contrastive loss. Another work, CrowdCLIP (Liang et al., 2023), focuses on the crowd counting problem, fine-tuning CLIP in an unsupervised manner to map crowd patches to count text. Concurrent research has focused on enabling VLM-driven image generation models to produce images with accurate item counts (Paiss et al., 2023; Li et al., 2023b; Kang et al., 2023).

Table 1: **The counting evaluation of the fine-tuned CLIP-Count model varies across diverse objects.** We use a CLIP-Count model checkpoint following its Visual Prompt Tuning and evaluate the Mean Absolute Error (MAE) and Root Mean Squared Error (RMSE) on our benchmark dataset `ObjectCount`. For comparison, we also compute MAE and RMSE for the pretrained `CLIP-base-32` model. The results indicate that CLIP-Count struggles with small-count tasks, performing poorly and showing a significant gap compared to the pretrained CLIP model.

|  |  | dogs | cats | lions | chairs | goats | cows | cherries | roses | boats |
|---|---|---|---|---|---|---|---|---|---|---|
| CLIP-Count | MAE | 9.87 | 50.02 | 16.59 | 28.59 | 35.56 | 34.16 | 13.01 | 25.53 | 56.32 |
|  | RMSE | 10.98 | 60.02 | 22.55 | 54.95 | 38.17 | 38.17 | 19.01 | 28.93 | 106.2 |
| CLIP-base-32 | MAE | 0.29 | 0.41 | 0.78 | 0.77 | 1.25 | 0.52 | 0.58 | 0.94 | 0.75 |
|  | RMSE | 0.58 | 0.70 | 1.16 | 1.06 | 1.60 | 0.76 | 0.89 | 1.23 | 1.16 |

**Density estimation framework vs. classification-based approaches in object counting**  Open-world counting tasks are often addressed using density estimation frameworks, which treat counting as a continuous regression problem, as demonstrated by CLIP-Count (Jiang et al., 2023b). In these methods, the model predicts a density map over an image, with performance typically evaluated using mean square error (MSE). This framework works well for large-scale object counting in high-density scenarios, where continuous approximations are often sufficient. This approach optimally suits high-count datasets, where density maps and MSE are effective evaluators of performance.

In contrast, classification-based approaches like ours formulate counting as a discrete classification task. Here, each potential object count is treated as a separate class, allowing the model to explicitly predict specific, integer-based counts. This task formulation is particularly well-suited for low-count scenarios, where the focus is on discrete accuracy rather than a smooth approximation.

While density estimation frameworks are powerful in high-density, large-count settings, they do not effectively capture the model's reasoning abilities in low-count scenarios, as shown in Table 1 Classification-based approaches offer a more nuanced assessment, emphasizing object type recognition and specific count accuracy. Thus, our work extends prior research by highlighting the distinct advantages of classification frameworks for low-count precision and reasoning in vision-language models.

**Linear word analogies and embedding editing**  Since word2vec (Mikolov et al., 2013a) was developed, researchers have found that the differences between word embedding vectors could capture relationships between words (Mikolov et al., 2013b; Drozd et al., 2016; Ethayarajh et al., 2018; Allen & Hospedales, 2019). For instance, the vector direction from "queen" to "king" captures a semantic transition from female to male. Building on this foundation, researchers have applied text embedding editing techniques to the field of image editing. Two works have explored the application of text embedding editing methods to image editing. One work (Parmar et al., 2023) discovers editing directions in the text embedding space and applies them to image edits, while leveraging cross-attention guidance to preserve the structure of image content. Another work (Nguyen et al., 2023) uses example pairs of 'before' and 'after' edits to infer text-based editing directions, which are then applied to new images, similar to (Parmar et al., 2023). Our research makes distinct contributions: (i) Using orthogonal projections, we filter extraneous details to achieve precise text embedding edits; (ii) Instead of focusing solely on image editing, we transfer CLIP's counting ability across objects to improve counting-related tasks, such as classification and generation.

# 3 Methods

In this section, we outline our approach to investigating and validating the hypothesis that the knowledge related to counting in images can be represented in a direction in the text embedding space such that it is independent of the object's embedding. Our approach enhances CLIP's counting abilities through learning-based methods that leverage new datasets and zero-shot text embedding editing techniques. We introduce the representation of counting concepts as vectors in Section 3.3, our parameter-efficient fine-tuning method in Section 3.4, and our zero-shot methods in Section 3.5. In Section 3.6, we will introduce how we collect and apply new counting datasets. Prior to the main method sections, we will first give a brief overview of CLIP models in Section 3.1 and define the counting problem and how to evaluate CLIP's counting ability in the following Section 3.2.

## 3.1 Overview of CLIP

Contrastive Language-Image Pre-Training (CLIP) (Radford et al., 2021) is a pioneering model developed by OpenAI that effectively integrates visual and textual data processing. The fundamental concept of CLIP involves the simultaneous training of two distinct encoders: an image encoder and a text encoder. These encoders are designed to produce embeddings that are closely aligned for corresponding image-text pairs and distinct for non-matching pairs.

The primary objective of CLIP is to minimize a contrastive loss that encompasses both image-to-text and text-to-image directions. Specifically, for a given dataset with $N$ samples, where $\mu_i \in R^d$ denotes the normalized image embedding and $v_i$ denotes the normalized text embedding for the $i^{\text{th}}$ sample, the CLIP loss function is defined as:

$$L_{\text{CLIP}} = -\frac{1}{2N} \sum_{i=1}^{N} \log \left( \frac{\exp(\mu_i \cdot v_i)}{\sum_{k=1}^{N} \exp(\mu_k \cdot v_i)} \right) - \frac{1}{2N} \sum_{i=1}^{N} \log \left( \frac{\exp(\mu_i \cdot v_i)}{\sum_{k=1}^{N} \exp(\mu_i \cdot v_k)} \right), \tag{1}$$

where we use $\cdot$ to denote the dot product between two vectors.

CLIP models are pre-trained on a large-scale dataset comprising 400 million image-text pairs sourced from the Internet. This extensive dataset enables CLIP to generalize well across different types of images and text found in real-world scenarios.

## 3.2 Evaluation of CLIP's counting accuracy

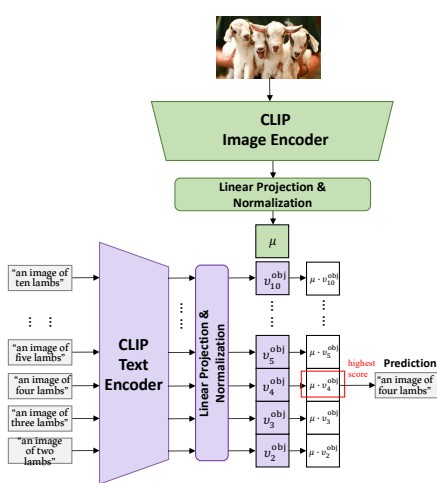

Figure 1: An illustration of object counting as a classification task with CLIP.

We start by defining the necessary notation. Let $v^{\text{obj}} \in \mathbb{R}^d$ be the CLIP text embedding vector for a caption like "an image of dogs", which identifies a specific object without indicating quantity, where $d$ is the embedding dimension of a CLIP model. $v_i^{\text{obj}} \in \mathbb{R}^d$ represents the text embedding of a caption that includes a quantifier, where $i$ is the quantity. For example, the embedding of the text "an image of *three* dogs" could be represented by $v_3^{\text{dog}}$.

To assess CLIP's counting performance, we set up an image classification task where the goal is to find a correct caption that describes the object count correctly in a given image, as illustrated in Figure 1.

Following (Paiss et al., 2023), we consider counts from two to ten as a nine-class classification task. Specifically, given an image with $n$ specific objects and nine candidate captions (e.g., "an image of $i$ objects" for $i$ between two and ten), we first encode the image with CLIP's image encoder to $\mu^{\text{obj}}$ and each caption with CLIP text encoder to $v_i^{\text{obj}}$. The cosine similarity

between the $\mu^{\mathrm{obj}}$ and each caption's text embedding $v_i^{\mathrm{obj}}$ is computed and used to select the caption yielding the highest similarity score.

### 3.3 Representation of counting knowledge as vectors

We define the representation of counting knowledge for each number as a vector $\Delta_i \in \mathbb{R}^d$, aligned with the dimensionality of CLIP's embedding space. Therefore, for the nine-class classification task that we consider, there is a set of 9 vectors representing different counts, each being denoted as $\Delta_i \in \mathbb{R}^d$ for $i \in \{2, 3, ..., 10\}$.

We then process these vectors, obtaining count vectors that are orthogonal to $v^{\mathrm{obj}}$, to eliminate information associated with object representation yet not contributing to object count. Accordingly, we introduce $\widetilde{\Delta}_i$ to denote the part of $\Delta_i$ that is orthogonal to $v^{\mathrm{obj}}$, as demonstrated in the top figure in Figure 2.

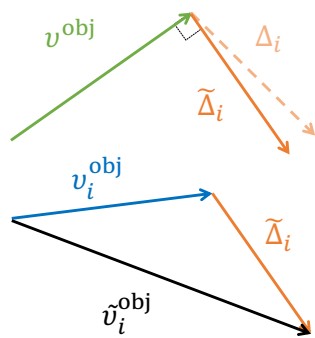

Then, we derive a counting-augmented object representation $\widetilde{v}_i^{\mathrm{obj}}$ from original representation $v_i^{\mathrm{obj}}$ and the orthogonalized counting representation $\widetilde{\Delta}_i$, such that

$$\widetilde{v}_i^{\mathrm{obj}} = v_i^{\mathrm{obj}} + \widetilde{\Delta}_i, \tag{2}$$

where $\widetilde{\Delta}_i := \Delta_i - \frac{\Delta_i \cdot v^{\mathrm{obj}}}{v^{\mathrm{obj}} \cdot v^{\mathrm{obj}}} v^{\mathrm{obj}}$.

Note that our method only manipulates CLIP's text embedding and keeps its image embedding unchanged. The cosine similarity score between the original image embedding $\mu^{\mathrm{obj}}$ and each manipulated text embedding $\widetilde{v}_i^{\mathrm{obj}}$ is calculated to determine the object count in the caption with the highest similarity score.

Figure 2: Representing counting knowledge as vectors.

The choice of forcing $\Delta_i$ to be orthogonal to $v^{\mathrm{obj}}$ instead of to $v_i^{\mathrm{obj}}$ is based on empirical results, elaborated in the ablation study Section 5.2. We hypothesize that $v_i^{\mathrm{obj}}$ already contains some level of counting information. Thus, if we let $\Delta_i$ to be orthogonal to $v_i^{\mathrm{obj}}$, there might be some loss of counting information. Similarly, the choice of $v_i^{\mathrm{obj}}$ instead of $v^{\mathrm{obj}}$ in Equation 2 is also based on empirical studies. We also hypothesize that the existed counting information in $v_i^{\mathrm{obj}}$ will reinforce the counting signal in $\widetilde{v}_i^{\mathrm{obj}}$.

### 3.4 Learning counting knowledge vectors via counting loss

In this section, we introduce a learning-based method, also demonstrated in Figure 3, to obtain counting representation vectors $\widetilde{\Delta}_i$, by minimizing a counting-specific loss defined in the `CountBench` paper (Paiss et al., 2023). Specifically, given a pre-trained CLIP model, we freeze all its pre-trained weights and optimize only $\Delta_i$ for $i \in \{2, 3, ..., 10\}$, which has only $9d$ parameters.

To prepare for training, for each ground truth image-caption pair within the training dataset, we create eight counterfactual caption variants by manipulating only the object count in the original caption. For instance, if the correct caption is "two dogs," counterfactual variants include "three dogs," "four dogs," ..., "ten dogs." Our objective is to enhance similarity scores between the text-image pairs with correct counts compared to the counterfactual pairs, thereby improving counting accuracy. The counting loss $L_{\mathrm{count}}$ is defined by Paiss et al. (2023) as follows:

$$L_{\mathrm{count}} = -\frac{1}{N} \sum_{k=1}^{N} \sum_{j=2, j \neq t}^{10} \log \left( \frac{\exp(\mu^k \cdot \widetilde{v}_t^{\,k})}{\exp(\mu^k \cdot \widetilde{v}_t^{\,k}) + \exp(\mu^k \cdot \widetilde{v}_j^{\,k})} \right) \tag{3}$$

where $\mu^k$ is the normalized image embedding of the $k^{\mathrm{th}}$ sample in one batch, $\widetilde{v}_t^{\,k}$ is the normalized text embedding of a caption containing ground truth count $t$ of sample $k$, and $v_j^k$ represents the embeddings for a counting-specific counterfactual text that contains the wrong count $j$.

In the original paper (Paiss et al., 2023), where the authors continue training the CLIP model on a large amount of counting data of 158K images, they also include CLIP's regular pre-training contrastive loss in

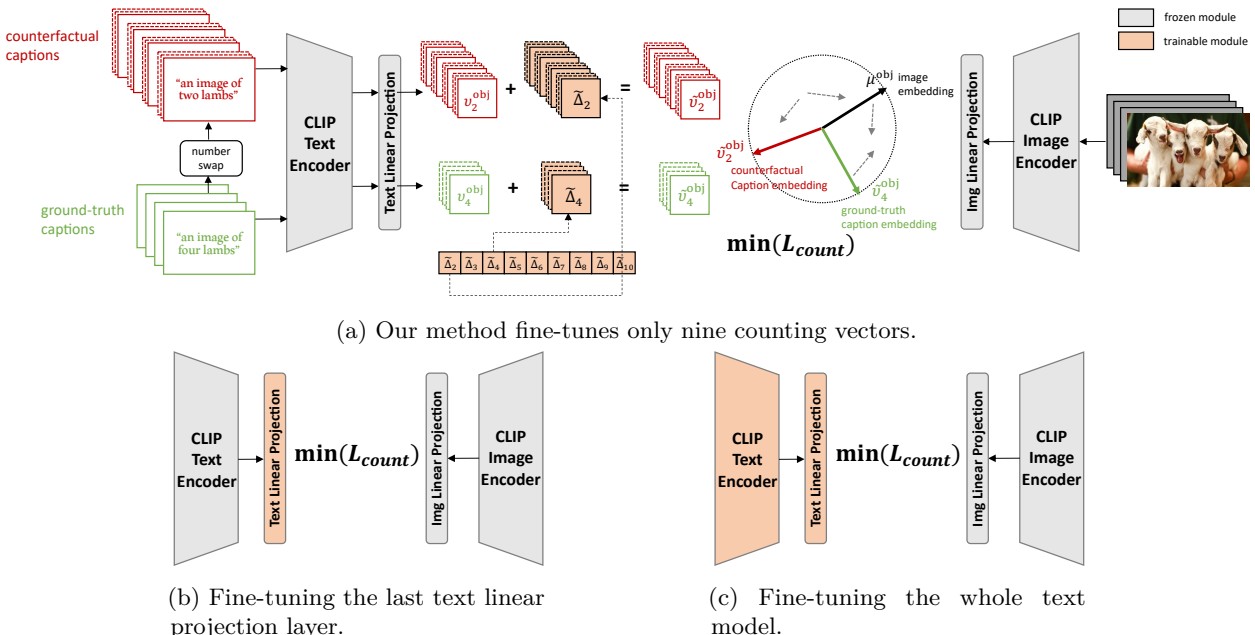

(a) Our method fine-tunes only nine counting vectors.

(b) Fine-tuning the last text linear projection layer.

(c) Fine-tuning the whole text model.

Figure 3: **Illustration of our learning-based approach.** As in Figure 3a, we follow Paiss et al. (2023) to generate counterfactual captions by swapping only the number word in each ground-truth caption, and fine-tune only nine counting vectors $\Delta_i$ as defined in Section 3.3 to minimize a counting loss $L_{\text{count}}$ that forces image embeddings to be far from counterfactual caption embedding yet close to ground-truth caption embeddings. We compare our special fine-tuning methods to two other fine-tuning methods shown in Figure 3b and Figure 3c, which involve training a larger number of model parameters.

the objective function $L$, such that $L = \lambda L_{\text{count}} + L_{\text{CLIP}}$. This is an explicit design to prevent CLIP from forgetting its other non-counting related pre-trained knowledge. We will also investigate the necessity of $L_{\text{CLIP}}$ in our setting where only a small amount of training data is available, in terms of its effectiveness in improving counting accuracy and preserving the CLIP model's pre-trained knowledge.

Recent work (Mestha et al., 2024) modifies $L_{\text{count}}$ to be an N-way contrastive loss (Sohn, 2016) and shows that it improves model performance after training more than using $L_{\text{count}}$ defined in Equation 3. We will detail the mathematical formula of the contrastive counting loss in our ablation study Section 5.1, and compare results after training models with the contrastive counting loss.

### 3.5 Zero-shot methods to transfer counting knowledge from prior knowledge

Alongside fine-tuning, we explore zero-shot methods to improve the CLIP model with counting capabilities without direct training on counting tasks. Zero-shot methods are valuable in addressing the scarcity of labeled image-text pairs for counting tasks, enhancing the model's practical applicability in real-world scenarios. We propose several techniques to extract counting knowledge representation vector $v_i$. These methods can also be used as initialization when training counting vectors, which can effectively speed up the optimization process discussed in the earlier section.

**Use text embedding of number words.** We encode each number $i$ in its English word with CLIP text encoder and denote it as $\Delta_i^{\text{num}}$. For example, the text embedding of word "two" is denoted as $\Delta_2^{\text{num}}$. Therefore, following Equation 2, the counting-augmented object representation is calculated as $\widetilde{v}_i^{\text{obj}} = v_i^{\text{obj}} + \widetilde{\Delta}_i^{\text{num}}$, where $\widetilde{\Delta}_i^{\text{num}}$ is $\Delta_2^{\text{num}}$ that's orthogonalized w.r.t $v^{\text{obj}}$.

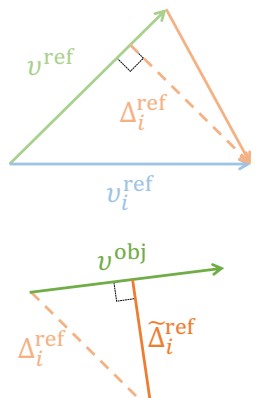

Figure 4: Extracting counting vector from a reference object.

**Extract counting knowledge from an easy-to-count object.** This approach is based on the observation that CLIP is more proficient at counting certain types of objects, such as dogs and cats, as elaborated in Section 4.4. Our key idea is that if CLIP effectively counts certain object types, it already possesses some counting knowledge, at least for those objects. When we have the prior knowledge of which object CLIP can count effectively, we take it as a counting reference and extract $\Delta_i$ from this object.

Specifically, as shown in Figure 4, we define the counting representation extracted from any reference object as $\Delta_i^{\text{ref}}$, such that

$$\Delta_i^{\text{ref}} = (v_i^{\text{ref}} - v^{\text{ref}}) - \frac{(v_i^{\text{ref}} - v^{\text{ref}}) \cdot v^{\text{ref}}}{v^{\text{ref}} \cdot v^{\text{ref}}} v^{\text{ref}}. \tag{4}$$

The intuition behind this definition is that the counting information is encapsulated in the direction moving from the non-quantitative representation ($v^{\text{ref}}$) to the quantitative representation ($v_i^{\text{ref}}$). Similarly, we obtain $\Delta_i^{\text{ref}}$ by making $v_i^{\text{ref}} - v^{\text{ref}}$ orthogonal to $v^{\text{ref}}$ to eliminate information associated with the non-quantitative representation. We further orthogonalize it to the non-quantitative representation of the target object before applying it. We denote the counting representation after two steps of orthogonalization as $\widetilde{\Delta}_i^{\text{ref}}$, such that

$$\widetilde{\Delta}_i^{\text{ref}} := \Delta_i^{\text{ref}} - \frac{\Delta_i^{\text{ref}} \cdot v^{\text{obj}}}{v^{\text{obj}} \cdot v^{\text{obj}}} v^{\text{obj}}. \tag{5}$$

Therefore, following Equation 2, the counting-augmented object representation is calculated as $\widetilde{v}_i^{\text{obj}} = v_i^{\text{obj}} + \widetilde{\Delta}_i^{\text{ref}}$.

**Extract knowledge from multiple objects.** Instead of relying on specific objects known to be easier for CLIP to count, this method aggregates counting vectors from a diverse set of objects, which aims to create a robust counting mechanism. We prompt ChatGPT (OpenAI, 2024), specifically `gpt-4-turbo-2024-04-09`, to generate a list of 100 common daily objects and animals in their plural form. For each object in the list, we calculate the counting-specific vector $\widetilde{\Delta}_i^{\text{ref}}$, same as described in the above paragraph. We then average these vectors across all objects in the list to create a generalized counting vector $\widetilde{\Delta}_i^{\text{multi}} = \frac{1}{100} \sum_{j=1}^{100} \widetilde{\Delta}_i^{\text{ref}_j}$. This averaged vector $\widetilde{\Delta}_i^{\text{multi}}$, is used as the counting representation to calculate $\widetilde{v}_i^{\text{obj}} = v_i^{\text{obj}} + \widetilde{\Delta}_i^{\text{multi}}$, hypothesizing that it encapsulates a universal counting pattern applicable across different object types.

### 3.6 Development of new object counting benchmarks

To rigorously evaluate the counting capabilities of the CLIP model and the effectiveness of our methods, we have developed two datasets in addition to using existing benchmark `CountBench` (Paiss et al., 2023). `CountBench` is an object counting dataset, collected from the LAION-400M dataset (Schuhmann et al., 2021). It comprises 540 images in total, with each numerical count represented by 60 respective images of different types of objects. Each image is accompanied by a caption describing the count of a specific object.

**A new diverse-source counting dataset.** The first new benchmark `DiverseCount` consists of images automatically sourced from multiple sources, including the COCO Dataset (Lin et al., 2014), Conceptual 12M (Changpinyo et al., 2021), YFCC100M (Thomee et al., 2016), and SBU Captions Dataset (Ordonez et al., 2011). In addition, due to the fact that images with large counts (i.e., nine and ten) are scarce, to compose 150 images for each count, we also manually collect some images from the Internet. Each text-image pair is manually reviewed, and captions are revised to eliminate grammatical errors and irrelevant information. Images are also manually checked to avoid duplication. This process yielded a more comprehensive set of images with clear, concise captions, containing 1350 images in total.

**An object-specific counting dataset.** The second new benchmark `ObjectCount` focuses on object-specific counting to delve deeper into how object types influence counting performance.

It includes 360 images in total, with 10 images for each of nine different objects $\in$ {"dog", "cats", "lion", "chair", "goat", "cow", "cherry", "rose", "boat"}, at each count level from two to five, all manually collected from the Internet. Counts are capped at five due to the scarcity of images with higher object counts. This dataset is used to evaluate how counting ability varies across different object types and to identify objects that are easier to count, which can serve as references in zero-shot methods.

# 4 Experiments and Results

## 4.1 Experimental setup

**Models.** We evaluate our method on three CLIP variants: `clip-vit-base-patch32`, `clip-vit-base-patch16`, and `clip-vit-large-patch14`. These models have progressively smaller patch sizes, allowing each model to capture finer image details. Furthermore, `clip-vit-large-patch14` has a larger model size compared to the first two models. We test our zero-shot methods on the Stable Diffusion model, `stable-diffusion-v1-4`, sourced from HuggingFace.

**Learning-based methods implementation details.** We compared our novel approach, which involves optimizing counting vectors, with two conventional fine-tuning methods: 1) full fine-tuning of the pre-trained CLIP text encoder and 2) fine-tuning only pre-trained CLIP's last linear text projection layer, 3) prefix-tuning with prefix length 5 (Li & Liang, 2021). Additionally, we test whether including the standard CLIP contrastive loss ($L_{\mathrm{CLIP}}$) enhances performance on small-scale datasets. For training the counting vectors, we set a higher learning rate of $10^{-3}$. We use lower learning rates of $10^{-4}$ for fine-tuning the text projection layer and $10^{-5}$ for the entire text model to minimize overfitting.

We divide our new dataset, `DiverseCount`, into training, validation, and test sets in a 6:2:2 ratio. We run three experiments with different random seeds and report the average test set scores to ensure robustness. We track performance across epochs by saving checkpoints and selecting the one with the lowest validation loss for final evaluations. We first evaluate each model on the `DiverseCount` test set, followed by `CountBench` to measure generalization across data distributions.

To evaluate the impact of fine-tuning on unrelated tasks, we test the models onCIFAR10 (Krizhevsky et al., 2009), CIFAR100 (Krizhevsky et al., 2009), Caltech101 (Fei-Fei et al., 2004), EuroSAT (Helber et al., 2018; 2019), and Food101 (Bossard et al., 2014). This assessment helps us evaluate the role of $L_{\mathrm{CLIP}}$ in preserving CLIP's pre-trained capabilities.

**Zero-shot methods implementation details.** For zero-shot methods, we implemented different choices of counting vectors $\Delta_i$ introduced in Section 3.5, including: 1) text embedding of number words $\Delta_i^{\mathrm{num}}$; 2) counting vectors $\widetilde{\Delta}_i^{\mathrm{multi}}$ extracted from common objects; and 3) counting vectors $\widetilde{\Delta}_i^{\mathrm{ref}}$ extracted from an easy-to-count object. In the third set of experiments, we select 'cats' and 'dogs' as reference objects based on their strong performance in Table 4.

We evaluate our method on our custom dataset `ObjectCount`, as introduced in Section 3.6, as well as on the image counting benchmark, `CountBench` (Paiss et al., 2023). The counting task on `ObjectCount` has four classes (counts from two to five), while `CountBench` includes nine classes (counts from two to ten).

## 4.2 Effectiveness of learned counting knowledge vectors

We assess the effectiveness of our methods in improving CLIP's counting accuracy using the `DiverseCount` test set and `CountBench`. The results in Table 2a indicate that training counting vectors and fine-tuning the last linear layer are more effective than fine-tuning the entire text model, even with far fewer parameters. This supports our hypothesis that counting knowledge may be encapsulated in a specific direction applicable across different objects. Training counting vectors, involving only $9d$ parameters, is computationally efficient and maintains strong performance.

Table 2: **Accuracy (%) of 9-class classification on `DiverseCount` test splits and on `CountBench`, comparing different fine-tuning methods.** All models are trained on the `DiverseCount` training split. We report the average accuracy from three runs with different train/val/test splits. Columns under $L = L_{\text{count}}$ represent methods that optimize only the counting loss, while columns under $L = L_{\text{count}} + L_{\text{CLIP}}$ represent methods that optimize both the counting loss and CLIP's standard contrastive loss. "CntVecs," "Proj," "Text Model," and "Prefix Tuning" each indicate which parameter is exclusively trained: "counting vectors," "CLIP's text projection layer," "CLIP's text model," and "prefix embedding vectors," respectively. We boldface the highest score in each row.

(a) **Accuracy of 9-class classification on `DiverseCount` test splits, comparing different fine-tuning methods.**

| Model | Original | Learning-based methods | | | | | | | |
| | | $L = L_{\text{count}}$ | | | | $L = L_{\text{count}} + L_{\text{CLIP}}$ | | | |
| | | CntVecs (ours) | Proj | Text Model | Prefix Tuning | CntVecs (ours) | Proj | Text Model | Prefix Tuning |
| `CLIP-base-32` | 28.17 | 37.78 | 38.16 | 33.92 | 14.55 | **38.53** | 37.66 | 34.17 | 15.67 |
| `CLIP-base-16` | 28.66 | 38.28 | 38.53 | 35.41 | 13.81 | **38.66** | 38.03 | 36.04 | 17.91 |
| `CLIP-large-14` | 33.62 | **41.34** | 39.6 | 33.49 | 13.06 | 40.84 | 39.35 | 33.99 | 17.54 |

(b) **Accuracy of 9-class classification on `CountBench`, comparing different fine-tuning methods.**

| Model | Original | Learning-based methods | | | | | | | |
| | | $L = L_{\text{count}}$ | | | | $L = L_{\text{count}} + L_{\text{CLIP}}$ | | | |
| | | CntVecs (ours) | Proj | Text Model | Prefix Tuning | CntVecs (ours) | Proj | Text Model | Prefix Tunng |
| `CLIP-base-32` | 30.69 | 33.12 | 27.47 | 32.83 | 10.21 | 33.4 | 27.75 | **33.62** | 15.32 |
| `CLIP-base-16` | 28.73 | 30.66 | 27.39 | 28.51 | 15.32 | **30.89** | 27.25 | 29.47 | 18.51 |
| `CLIP-large-14` | 31.97 | 39.2 | 31.4 | 32.33 | 10.45 | **39.41** | 31.83 | 32.47 | 13.65 |

When evaluated on `CountBench` (as shown in Table 2b), all methods exhibit reduced effectiveness, likely due to a distribution gap between `DiverseCount` and `CountBench`. Training counting vectors continues to improve CLIP's counting accuracy on `CountBench`, demonstrating better generalization than other methods.

We also observe that prefix tuning, which fine-tunes a similar number of parameters, does not improve performance and negatively impacts counting accuracy.

Moreover, the inclusion of $L_{\text{CLIP}}$ in the training objective does not significantly influence the outcomes across all training setups, as seen in the three rightmost columns in Table 2.

### 4.3 Impact on CLIP's performance in non-counting Tasks

Fine-tuning via directly updating CLIP's pre-trained weight has a significant impact on CLIP's performance in non-counting benchmarks, as detailed in Table 3. Both fine-tuning the entire text model and the last linear layer results in noticeable performance drops across all benchmarks, with fine-tuning the projection layer having a more pronounced effect.

Incorporating $L_{\text{CLIP}}$ provides only marginal effectiveness in preventing the loss of pre-trained knowledge. This contrasts with the findings in the `CountBench` paper, where incorporating $L_{\text{CLIP}}$ was beneficial when fine-tuning CLIP with a large dataset (i.e., 158K images), providing ample sources for the model to learn new things. In our experiments, where only a smaller dataset was available, $L_{\text{CLIP}}$ did not demonstrate the same effectiveness, suggesting that its utility may be limited under conditions of restricted data availability.

However, training new counting vectors does not alter CLIP's model parameters. Thus, we can apply these vectors only in tasks specific to counting, while relying solely on the pre-trained CLIP model for all other non-counting tasks.

Table 3: **Performance on common benchmarks.** We compare the performance of fine-tuned models against pre-trained models on common non-counting benchmarks. Fine-tuning either linear layer or the whole model will lead to siginificant performance drop.

| | | | Learning-based methods | | | |
| | | | $L = L_{\text{count}}$ | | $L = L_{\text{count}} + L_{\text{CLIP}}$ | |
| Model | Benchmark | Original | Proj | Text Model | Proj | Text Model |
|---|---|---|---|---|---|---|
| | CIFAR10 | 88.95 | 83.43 | **89.01** | 82.72 | 88.95 |
| | CIFAR100 | **48.78** | 35.68 | 47.75 | 37.9 | 47.62 |
| CLIP-base-32 | Caltech101 | 80.18 | 71.50 | **80.33** | 71.15 | 80.3 |
| | EuroSAT | **45.11** | 36.49 | 43.78 | 37.94 | 44.28 |
| | Food101 | **80.2** | 78.64 | 78.61 | 78.59 | 78.69 |
| | CIFAR10 | 88.35 | 79.92 | 88.31 | 81.59 | **88.38** |
| | CIFAR100 | 58.63 | 51.98 | 58.95 | 53.38 | **59.36** |
| CLIP-base-16 | Caltech101 | **76.78** | 70.70 | 75.13 | 71.18 | 75.67 |
| | EuroSAT | **49.83** | 39.20 | 46.84 | 43.77 | 47.47 |
| | Food101 | **85.57** | 82.97 | 85.26 | 85.69 | 83.49 |
| | CIFAR10 | 95.01 | 93.65 | 94.91 | 93.71 | **95.04** |
| | CIFAR100 | 64.66 | 62.49 | 64.9 | 62.47 | **64.99** |
| CLIP-large-14 | Caltech101 | **81.02** | 75.50 | 80.20 | 75.71 | 80.56 |
| | EuroSAT | **55.37** | 51.59 | 54.00 | 51.46 | 53.9 |
| | Food101 | **89.79** | 88.46 | 89.35 | 88.61 | 89.43 |

Table 4: **The counting accuracy of CLIP varies across diverse objects.** Pre-trained CLIP models counting accuracy varies by object type. Still, all models consistently count dogs and cats more accurate than other objects.

| | average | dogs | cats | lions | chairs | goats | cows | cherries | roses | boats |
|---|---|---|---|---|---|---|---|---|---|---|
| CLIP-base-32 | 47.93 | **58.86** | **66.14** | 47.73 | 35.23 | 42.73 | 46.36 | 45.45 | 32.27 | 47.27 |
| CLIP-base-16 | 50.33 | **74.77** | **74.77** | 54.32 | 47.05 | 32.73 | 55.00 | 35.00 | 34.09 | 45.23 |
| CLIP-large-14 | 60.86 | **75.23** | **79.09** | 65.45 | 52.95 | 44.77 | 65.00 | 53.86 | 56.82 | 54.55 |

## 4.4 CLIP's counting ability on different objects

Each column in Table 4 represents the counting accuracy for a specific object in `ObjectCount`, with object names as column headers. The average accuracy across all objects is also displayed under the "average" column. We observe a positive correlation between model size and average counting accuracy, with accuracies ranging from 47.93% to 60.86%. However, there is significant variation in the models' counting abilities for different object types, suggesting that CLIP's counting capability is dependent on the object.

Notably, all models consistently perform best when counting "dogs" and "cats", while their performance with other objects lacks consistency. We hypothesize that the superior performance on "dogs" and "cats" may result from their higher frequency in the pre-training dataset. In fact, when collecting our dataset `ObjectCount` from the Internet, we do observe that images of dogs and cats are more accessible in larger volumes than images of other objects.

## 4.5 Effectiveness of our zero-shot method

We evaluate our zero-shot methods on our custom dataset `ObjectCount` and `CountBench`, and report the results in Table 5. Across all three CLIP models, there are noticeable improvements in counting accuracy when using zero-shot methods compared to the default (baseline) settings. However, the effectiveness of each strategy varies by task and model size. For example, most methods are more effective on the `CLIP-large-14`

Table 5: **CLIP's counting accuracy for image classification task on our custom dataset `ObjectCount` and `CountBench` (%), comparing results of zero-shot methods.** We bold the higest score in each row.

| Model | DatasetMethod | original | $v_i^{\text{obj}} + \widetilde{\Delta}_i^{\text{multi}}$ | $v_i^{\text{obj}} + v_i^{\text{num}}$ | $v_i^{\text{obj}} + \widetilde{\Delta}_i^{\text{dogs}}$ | $v_i^{\text{obj}} + \widetilde{\Delta}_i^{\text{cats}}$ |
|---|---|---|---|---|---|---|
| `CLIP-base-32` | `ObjectCount` (4-class) | 47.93 | 49.65 | 49.65 | **51.84** | 49.8 |
| | `CountBench` (9-class) | 30.69 | **31.91** | 31.7 | 28.3 | 29.36 |
| `CLIP-base-16` | `ObjectCount` (4-class) | 50.33 | 53.16 | 52.55 | **55.45** | 54.77 |
| | `CountBench` (9-class) | 28.73 | 27.02 | **31.06** | 29.15 | 28.51 |
| `CLIP-large-14` | `ObjectCount` (4-class) | 60.86 | **65.58** | 64.77 | 64.12 | 64.5 |
| | `CountBench` (9-class) | 31.97 | **39.45** | 36.25 | 39.23 | 38.17 |

model, which has larger model size and higher resolution, with improvement close to or higher than 5%, The improvement is more significant on `ObjectCount` than on `CountBench`, likely because `CountBench` includes nine classes, making it more challenging.

## 4.6 Effectiveness of our method in improving text-to-image models' counting fidelity

Since our method directly enhances the counting capability of the CLIP model in a zero-shot manner, we anticipate that using our approach will also aid models that utilize CLIP embeddings for image generation, such as the Stable Diffusion model (Rombach et al., 2021), in producing images with the correct number of objects. Therefore, we experiment with applying our method to Stable Diffusion. We select 10 image descriptions from `CountBench` (Paiss et al., 2023), as shown in Table 6. Each description features a commonly counted object and can serve as an appropriate image generation prompt for Stable Diffusion. We modify the counting number in each description to range from "two" to "ten," forming 90 prompts in total, each describing a unique combination of counting number and object. These prompts are provided to the Stable Diffusion model[1], and we generate 20 images for each prompt based on both our method and the original CLIP model, resulting in a total of 3,600 images. We use the YOLOv9 model (Wang & Liao, 2024) for object detection to compare the object counts in images generated by our method and the original CLIP model. This allows us to determine how many instances of the specified object are present, using the detection results as ground truth.

Table 6: **List of prompts used for image generation, with the counting numbers and objects highlighted in blue and red, respectively.** For image generation, the counting numbers in each prompt are replaced with "two" to "ten".

| No. | Prompt |
|---|---|
| 1 | An old building with ruined walls and four antique pink armchairs |
| 2 | Vintage silver plate tablespoons, serving spoon set of two 1847 Rogers Ambassador pattern |
| 3 | Forks with vegetables—four forks with different types of... |
| 4 | Set of four stemless red wine glasses |
| 5 | Row of five British Shorthair cats sitting on a wooden tray, isolated on a white background |
| 6 | Eight bottles of aguardiente on a counter |
| 7 | Set of four multicolored 'Penzance' small bowls |
| 8 | Photo of ten giraffe portraits, isolated on a white background |
| 9 | Meet the MINI family, five cars in different styles, lined up in a row |
| 10 | Photo of seven red plastic apples on a white background |

The experimental results are presented in Figure 5, which displays the counting accuracies of images generated using the original CLIP model and our method, respectively, in the form of confusion matrices. We aggregate the image generation results for the ten types of objects. In the matrices, rows represent the specified counts

---

[1] https://huggingface.co/CompVis/stable-diffusion-v1-4

in prompts, and columns represent the detected counts in generated images. The red boxes highlight the cells where the detected number of objects matches the counting number specified in the prompt, indicating the number of images correctly generated for each counting number. For prompts with counts from 2 to 5, our method significantly improves Stable Diffusion's accuracy in generating images with the correct object counts. However, for counting numbers from 6 to 10, the effectiveness of our method diminishes, likely because the CLIP model's pretrained dataset lacks text labels corresponding to these higher counting numbers. Examples of images generated by Stable Diffusion are shown in Table 7 and Appendix A. Note that our method can be used in conjunction with existing methods for improving the fidelity of text-to-image models, e.g., reinforcement learning-based algorithms (Fan et al., 2023; Fan & Lee, 2023; Black et al., 2023).

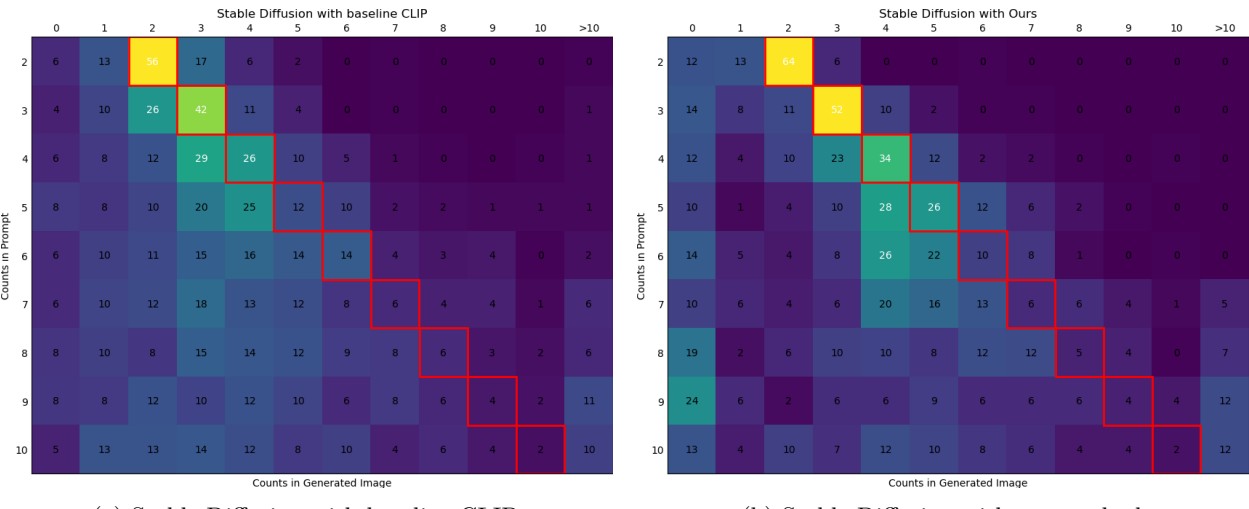

(a) Stable Diffusion with baseline CLIP  (b) Stable Diffusion with our method

Figure 5: **Confusion matrices comparing object counting accuracy in images generated by Stable Diffusion using (a) baseline CLIP and (b) our method.** In each matrix, the rows represent the counts specified in the prompts for image generation, and the columns represent the counts detected in the generated images using the corresponding prompts. The red boxes highlight the cells where the detected number of objects matches the count specified in the prompt, indicating the ratio of generated images with correct counts for each row. We observe that our method is more effective in helping the Stable Diffusion model generate images with correct counts between 2 and 5, as shown by higher numbers in the red boxes, but less effective for counts from 6 to 10.

## 5 Ablation Studies

### 5.1 Effectiveness of different contrastive loss designs

In our work, we adapt the count loss proposed by Paiss et al. (2023), as shown in Equation 3. We compare it against a multi-class N-way loss, denoted by $\widetilde{L}_{\text{count}}$, used in Mestha et al. (2024), which is defined as:

$$\widetilde{L}_{\text{count}} = -\frac{1}{N}\sum_{k=1}^{N}\log\left(\frac{\exp(\mu^k \cdot \widetilde{\upsilon}_t^{\,k})}{\exp(\mu^k \cdot \widetilde{\upsilon}_t^{\,k}) + \sum_{j=2, j\neq t}^{10}\exp(\mu^k \cdot \widetilde{\upsilon}_j^{\,k})}\right) \tag{6}$$

Table 8 shows that on the `DiverseCount` test split, fine-tuning with the contrastive counting loss $\widetilde{L}$count performs slightly worse than $L$count.. However, applying $\widetilde{L}_{\text{count}}$ further improves the accuracy of most fine-tuned models on the `CountBench` dataset, indicating that $\widetilde{L}_{\text{count}}$ might have a more robust and effective loss design when the models need to generalize to setting with larger distribution shift.

Table 7: **Selected results from Stable Diffusion (Rombach et al., 2021).** Images in the "Original" column are generated based on the input prompt in the same row, using different seeds. Images in the "Embedding edited" column are generated after applying our zero-shot method (using the same seeds), with the selection of "dog" as the reference. After applying our method, we observe that Stable Diffusion is more likely to generate images with the correct number of objects.

| Input Prompt | Original | Embedding edited |
|---|---|---|
| **"three** lions" |  |  |
| "An old building with ruined walls and **four** antique pink armchairs" |  |  |
| "vintage silver plate tablespoons, serving spoon set of **two**" |  |  |
| **"three** dolphins jumping out of water" |  |  |
| "A picture of **three** cherries" |  |  |

Table 8: **Accuracy (%) of 9-class classification on `DiverseCount` test splits and `CountBench` of different fine-tuning methods, with contrastive count loss $\widetilde{L}_{\mathbf{count}}$.** For each fine-tuning method, we bold scores if its higher when applying contrastive loss $\widetilde{L}_{\text{count}}$ vs. $L_{\text{count}}$. We display changes compared to applying $L_{\text{count}}$ inside the parenthesizes right after each score.

| Model | DiverseCount | | | CountBench | | |
|---|---|---|---|---|---|---|
| | CntVecs (ours) | Proj | Text Model | CntVecs (ours) | Proj | Text Model |
| `CLIP-base-32` | 37.28 (-0.50) | 37.78 (-0.38) | 33.78 (-0.14) | **33.55 (+0.43)** | **28.25 (+0.78)** | **34.62 (+1.79)** |
| `CLIP-base-16` | 38.03 (-0.25) | 38.03 (-0.5) | **35.53 (+0.12)** | **32.66 (+2.00)** | **28.66 (+1.27)** | **29.4 (+1.89)** |
| `CLIP-large-14` | 39.48 (-1.86) | 39.59 (-0.01) | **34.37 (+0.88)** | 37.91 (-1.29) | **32.69 (+1.29)** | **33.48 (+1.15)** |

## 5.2 Ablation studies of each component in the counting representation

### 5.2.1 Learning-based settings

To evaluate the impact of different design choices in learning-based settings, we analyze modifications in representing and applying $\widetilde{\Delta}_i$. In our main method, $\widetilde{\Delta}_i := \Delta_i - \frac{\Delta_i \cdot v^{\text{obj}}}{v^{\text{obj}} \cdot v^{\text{obj}}} v^{\text{obj}}$, which involves orthogonalization with respect to $v^{\text{obj}}$. We study the effect of orthogonalization in the `NoObjOrth` experiment. We also study the impact of the projection direction used during orthogonalization, by comparing the main results to the design such that $\Delta_i$ is forced to be orthogonal to $v_i^{\text{obj}}$ instead of $v^{\text{obj}}$. We name this group of experiments, where $\widetilde{\Delta}_i := \Delta_i - \frac{\Delta_i \cdot v^{\text{obj}}}{v^{\text{obj}} \cdot v^{\text{obj}}}$, as in `ChangeProjDir`. In addition, when applying $\widetilde{\Delta}_i$, it is added to $v_i^{\text{obj}}$, as shown in Equation 2. We study the effect of adding $\widetilde{\Delta}_i$ onto $v^{\text{obj}}$ such that $\widetilde{v}_i^{\text{obj}} = v^{\text{obj}} + \widetilde{\Delta}_i$, and named this group of experiments as `ChangeAddDir`.

Ablation study results are presented in Table 9, under columns corresponding to each ablation experiment. We find that in learning-based settings, there is no significant difference between each modification and the main approach. This suggests that when the same modifications are applied during training and inference,

the orthogonalization method and direction might not significantly affect the learning of counting vectors. However, these factors have much more significant effects in zero-shot settings, as we will demonstrate in the next section.

Table 9: **Accuracy (%) of 9-class classification on `DiverseCount` test splits and `CountBench` of fine-tuning counting vectors, comparing different modifications with our method in the main paper.** For each modification, we bold scores if its higher than score of the main method.

| Model | DiverseCount | | | | CountBench | | | |
|---|---|---|---|---|---|---|---|---|
| | Main method | NoObjOrth | ChangeProjDir | ChangeAddDir | Main method | NoObjOrth | ChangeProjDir | ChangeAddDir |
| CLIP-base-32 | 37.78 | 37.16 | 37.66 | 36.28 | 33.12 | **33.62** | **33.26** | 32.69 |
| CLIP-base-16 | 38.28 | **38.65** | **38.78** | 38.03 | 30.66 | **31.7** | **31.48** | **32.22** |
| CLIP-large-14 | 41.34 | **42.59** | 41.09 | **41.84** | 39.2 | **39.41** | **39.34** | **40.27** |

### 5.2.2 Zero-shot settings

In addition to ablation study groups introduced in the previous section, including `NoObjOrth`, `ChangeProjDir`, `ChangeAddDir`, for zero-shot settings, we add another ablation study group named `NoRefOrth`. This ablation group studies the effect of orthogonalization w.r.t. reference object's text embeddings, by excluding projection on ref as defined in Equation 5, so that the orthogonalization only applies w.r.t. to target object's text embeddings $v^{\text{obj}}$ and that $\widetilde{\Delta}_i^{\text{ref}} = \Delta_i^{\text{ref}} - \frac{\Delta_i^{\text{ref}} \cdot v^{\text{obj}}}{v^{\text{obj}} \cdot v^{\text{obj}}} v^{\text{obj}}$.

Figure 6 and Table 10 demonstrate that the main method outperforms alternative modifications, particularly on the `CountBench` dataset.

### 5.3 Applications to other models

To evaluate the generalizability of our methods to larger, more recent models, we apply our counting-specific training techniques to the SigLIP (Zhai et al., 2023) and BLIP (Li et al., 2022) models. SigLIP is an alternative to CLIP, pre-trained using a simple pairwise sigmoid loss instead of CLIP's original loss function. The pairwise sigmoid loss improves SigLIP's zero-shot classification accuracy on ImageNet. BLIP, on the other hand, leverages bootstrapped self-supervision techniques that iteratively refine its understanding of images and text without extensive labeled data. This approach enables BLIP to excel in scenarios where labeled data may be sparse or unavailable, making it well-suited for open-world applications and robust zero-shot learning.

Our zero-shot method is evaluated on both SigLIP and BLIP. As shown in Table 11, our approach effectively enhances counting performance for these models under certain conditions. However, the improvements are not uniformly observed across all scenarios, indicating that our method's benefits may be context-dependent. Future research should focus on refining and extending our approach to ensure broader applicability across diverse models and datasets.

### 5.4 Implications for broader applicability

Although our findings with SigLIP are preliminary, they suggest that the observed gains are not restricted to smaller CLIP models but may extend to larger, state-of-the-art architectures. Fine-tuning the text embedding space to incorporate numerical representations may be an efficient strategy for enhancing counting accuracy across diverse vision–language models. Future work could explore this approach in additional high-capacity models and evaluate it on more diverse counting benchmarks, providing deeper insights into its adaptability and robustness.

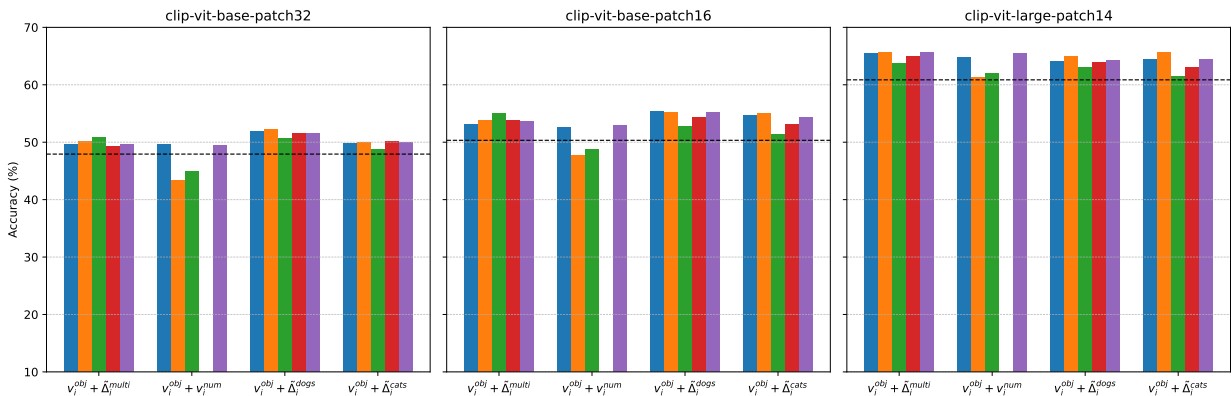

(a) Ablation studies of zero-shot methods on `ObjectCount` dataset.

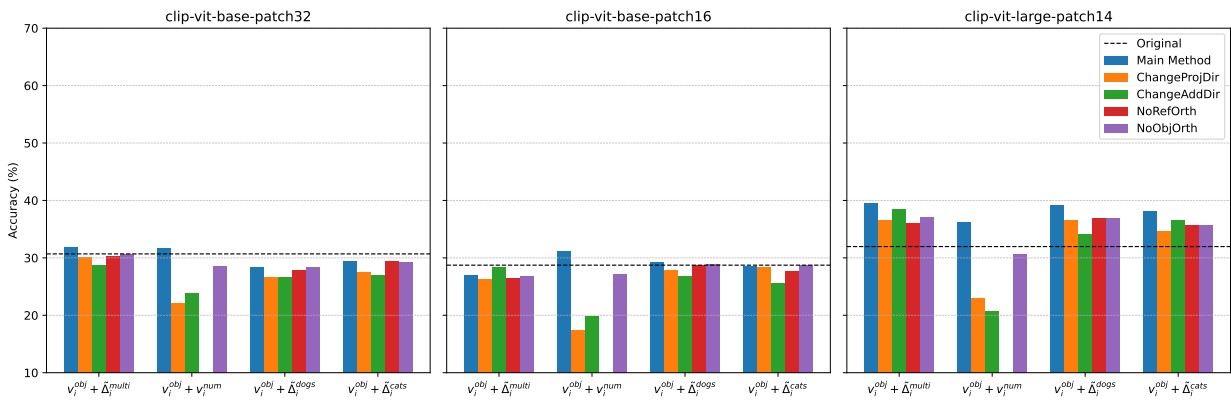

(b) Ablation studies of zero-shot methods on `CountBench` dataset.

Figure 6: Ablation studies of zero-shot methods on `ObjectCount` dataset and `CountBench` dataset.

# 6    Conclusion and Discussion

## 6.1    Summary of contributions

In this study, we explored the counting capabilities of CLIP models and introduced a computationally efficient method for training counting vectors, along with several zero-shot text embedding editing techniques to enhance CLIP's counting accuracy. Our learning-based approach demonstrates that targeted modifications to text embeddings can significantly improve object counting tasks without the need for extensive model retraining. This method not only proves to be effective but also preserves the broader capabilities of CLIP, unlike other methods that might compromise the general performance of CLIP models.

Our zero-shot techniques are particularly valuable in contexts where data is scarce or full model retraining is impractical due to computational constraints. These methods have also shown promise when applied to text-to-image models like Stable Diffusion, indicating their potential applicability beyond the initial use case.

## 6.2    Performance variation across object categories

Our analysis reveals that CLIP's counting accuracy varies significantly across different object categories, with a notable advantage for common objects such as cats and dogs. This outcome can be attributed to the pretraining data distribution of CLIP, where categories that are frequently represented in this dataset benefit from more comprehensive visual and linguistic representations, enabling more accurate counting results. In

Table 10: **CLIP's counting accuracy (%) on our custom `ObjectCount` and `CountBench` datasets for image classification tasks, comparing zero-shot methods.** For each zero-shot method, we bold scores that are higher under the current modification. Changes relative to the main method are shown in parentheses following each score.

(a) **CLIP's counting accuracy for image classification task on our custom dataset `ObjectCount`**

| | Model | $v_i^{\mathrm{obj}} + \widetilde{\Delta}_i^{\mathrm{multi}}$ | $v_i^{\mathrm{obj}} + v_i^{\mathrm{num}}$ | $v_i^{\mathrm{obj}} + \widetilde{\Delta}_i^{\mathrm{dogs}}$ | $v_i^{\mathrm{obj}} + \widetilde{\Delta}_i^{\mathrm{cats}}$ |
|---|---|---|---|---|---|
| Change Add Dir | `CLIP-base-32` | **50.81 (+1.16)** | 44.87 (-4.78) | 50.76 (-1.08) | 48.79 (-1.01) |
| | `CLIP-base-16` | **55.13 (+1.97)** | 48.79 (-3.76) | 52.75 (-2.7) | 51.39 (-3.38) |
| | `CLIP-large-14` | 63.79 (-1.79) | 61.94 (-2.83) | 63.13 (-0.99) | 61.47 (-3.03) |
| Change Proj Dir | `CLIP-base-32` | **50.18 (+0.53)** | 43.46 (-6.19) | **52.2 (+0.36)** | **50.08 (+0.28)** |
| | `CLIP-base-16` | **53.76 (+0.6)** | 47.68 (-4.87) | 55.25 (-0.2) | **55.08 (+0.31)** |
| | `CLIP-large-14` | **65.68 (+0.1)** | 61.39 (-3.38) | **65.05 (+0.93)** | **65.71 (+1.21)** |
| No Ref Orth | `CLIP-base-32` | 49.34 (-0.31) | - | 51.59 (-0.25) | **50.15 (+0.35)** |
| | `CLIP-base-16` | **53.81 (+0.65)** | - | 54.29 (-1.16) | 53.21 (-1.56) |
| | `CLIP-large-14` | 65.05 (-0.53) | - | 63.99 (-0.13) | 63.13 (-1.37) |
| No Target Orth | `CLIP-base-32` | **49.72 (+0.07)** | 49.47 (-0.18) | 51.56 (-0.28) | **49.93 (+0.13)** |
| | `CLIP-base-16` | **53.61 (+0.45)** | 53.05 (+0.5) | 55.25 (-0.2) | 54.42 (-0.35) |
| | `CLIP-large-14` | **65.71 (+0.13)** | **65.56 (+0.79)** | **64.24 (+0.12)** | 64.5 (+0.0) |

(b) **CLIP's counting accuracy for image classification task on our custom dataset `CountBench`**

| | Model | $v_i^{\mathrm{obj}} + \widetilde{\Delta}_i^{\mathrm{multi}}$ | $v_i^{\mathrm{obj}} + v_i^{\mathrm{num}}$ | $v_i^{\mathrm{obj}} + \widetilde{\Delta}_i^{\mathrm{dogs}}$ | $v_i^{\mathrm{obj}} + \widetilde{\Delta}_i^{\mathrm{cats}}$ |
|---|---|---|---|---|---|
| Change Add Dir | `CLIP-base-32` | 28.76 (-3.15) | 23.82 (-7.88) | 26.61 (-1.69) | 27.04 (-3.32) |
| | `CLIP-base-16` | **28.29 (+1.27)** | 19.82 (-11.24) | 26.73 (-2.42) | 25.61 (-2.9) |
| | `CLIP-large-14` | 38.41 (-1.04) | 20.6 (-15.65) | 34.12 (-5.11) | 36.48 (-1.69) |
| Change Proj Dir | `CLIP-base-32` | 30.04 (-1.87) | 22.1 (-9.6) | 26.61 (-1.69) | 27.47 (-1.89) |
| | `CLIP-base-16` | 26.28 (-0.74) | 17.37 (-13.69) | 27.84 (-1.31) | 28.29 (-0.22) |
| | `CLIP-large-14` | 36.48 (-2.97) | 22.96 (-13.29) | 36.48 (-2.75) | 34.55 (-3.62) |
| No Ref Orth | `CLIP-base-32` | 30.26 (-1.65) | - | 27.9 (-0.4) | **29.4 (+0.04)** |
| | `CLIP-base-16` | 26.5 (-0.52) | - | 28.73 (-0.42) | 27.62 (-0.89) |
| | `CLIP-large-14` | 36.05 (-3.4) | - | 36.91 (-2.32) | 35.62 (-2.55) |
| No Target Orth | `CLIP-base-32` | 30.69 (-1.22) | 28.54 (-3.16) | **28.33 (+0.03)** | 29.18 (-0.18) |
| | `CLIP-base-16` | 26.73 (-0.29) | 27.17 (-3.89) | 28.95 (-0.2) | **28.73 (+0.22)** |
| | `CLIP-large-14` | 37.12 (-2.33) | 30.69 (-5.56) | 36.91 (-2.32) | 35.62 (-2.55) |

contrast, less common or niche object categories that appear less frequently in the pretraining data exhibit reduced accuracy in counting tasks due to less robust feature learning.

This performance variability highlights a key challenge in open-world scenarios, where models must generalize effectively across both familiar and unfamiliar categories. While CLIP's broad pretraining endows it with impressive zero-shot learning capabilities, it is inherently limited by the distributional biases present in the dataset. As a result, when faced with objects outside the well-represented domain of its training corpus, CLIP may show diminished counting performance. This underscores the need for targeted fine-tuning or

Table 11: **Counting accuracy (%) of SigLIP and BLIP models on our custom `ObjectCount` and `CountBench` datasets for image classification tasks, comparing zero-shot methods.** The highest score in each row is shown in bold.

| Model | DatasetMethod | original | $v_i^{\text{obj}} + \widetilde{\Delta}_i^{\text{multi}}$ | $v_i^{\text{obj}} + v_i^{\text{num}}$ | $v_i^{\text{obj}} + \widetilde{\Delta}_i^{\text{dogs}}$ | $v_i^{\text{obj}} + \widetilde{\Delta}_i^{\text{cats}}$ |
|---|---|---|---|---|---|---|
| `siglip-base-patch16-224` | `ObjectCount` (4-class) | 22.12 | 25.82 | 26.40 | **26.84** | 25.39 |
| | `CountBench` (9-class) | **14.08** | 9.01 | 9.45 | 9.01 | 8.57 |
| `siglip-large-patch16-224` | `ObjectCount` (4-class) | 23.01 | 23.12 | **26.69** | 21.84 | 22.73 |
| | `CountBench` (9-class) | 12.31 | 12.53 | **14.51** | 12.31 | 10.33 |
| `blip-image-captioning-base` | `ObjectCount` (4-class) | 24.32 | 24.34 | 27.6 | **29.55** | 26.02 |
| | `CountBench` (9-class) | 10.94 | **14.00** | 12.74 | 11.82 | 9.63 |
| `blip-image-captioning-large` | `ObjectCount` (4-class) | **28.38** | 23.84 | 22.22 | 22.83 | 25.75 |
| | `CountBench` (9-class) | 12.91 | **17.72** | 9.41 | 10.94 | 11.82 |

adaptive strategies to bolster the model's counting capabilities for underrepresented or novel object types, enhancing its generalization in truly open-world contexts.

### 6.3 Image encoder assumptions and considerations

In this work, our method exclusively focused on the text encoder while assuming that the image encoder's existing capabilities were sufficient for counting. This choice was based on the hypothesis that CLIP's limitations in counting may be attributed to the representation of numerical concepts within its text embedding space. By enhancing this aspect, we demonstrated that counting performance could be improved without modifying the image encoder.

However, we acknowledge that the image encoder's limitations may also play a role in cases where counting performance is suboptimal. Factors such as feature extraction quality, spatial representation, and object separation within images can impact counting effectiveness. Investigating the image encoder's capabilities in relation to counting would be a valuable extension for future work, particularly to understand how joint adaptations of the image and text encoders might further improve counting tasks.

### 6.4 Count range and generalization limitations

Our study focused on a count range of 2 to 10, selected due to practical considerations such as dataset availability and the need for reliable image annotations. While our methods were shown to be effective within this range, we recognize that they may not generalize seamlessly to higher object counts without further adaptation. This is a limitation of our current approach, and future research could investigate strategies for expanding the count range, such as using hierarchical representations or synthetic data augmentation to improve generalization to larger counts.

### 6.5 Limitations and future work

Despite the promising results, our approach has inherent limitations. One key issue is the lack of a comprehensive theoretical understanding of why training counting-specific vectors outperforms other methods. Further exploration of the underlying mechanisms in CLIP's counting capabilities could provide new insights and inspire more robust enhancement strategies for vision-language models.

Additionally, our method is currently limited to scenarios involving a single counted quantity and does not support complex visual scenes with multiple object compositions or interactions between objects. For example, it cannot handle queries that require counting multiple object types simultaneously, such as "five dogs and three cats." Extending our method to handle multi-quantity counting would likely require more complex interactions within the text embedding space, and more advanced visual reasoning abilities into the model. Techniques such as compositional embedding strategies or multi-object relational modeling may provide pathways to expand the applicability of our approach to more complex counting queries.

Future work could focus on integrating spatial and relational reasoning into vision-language models to enhance their ability to interpret contextual and spatial cues within images, improving both counting accuracy and general visual comprehension. Extending our approach to other visual tasks, such as object detection and complex counting scenarios, could broaden its applicability and reveal universal strategies for boosting performance across various vision-language architectures.

Another limitation of this work is that our study focuses on low-count, object-specific scenarios and evaluates performance using the custom DiverseCount and ObjectCount datasets. These benchmarks were chosen to align with our problem formulation, which frames counting as a classification task. However, we recognize the importance of widely used benchmarks such as ShanghaiTech (Zhang et al., 2016) and UCF-QNRF(Idrees et al., 2018), which provide comprehensive evaluations of counting methods. Although these data sets are typically used in density estimation tasks, they can also be adapted for classification-based evaluation by aggregating object counts in defined regions or across the entire image.

Our current work does not include experiments on these benchmarks, which limits the scope of our evaluation. Future research should explore the application of our methods to such datasets, enabling a broader understanding of their effectiveness across diverse counting paradigms and further validating their generalizability.

**Acknowledgments**

This work is supported in part by NSF CAREER Award CCF-2339978, Amazon Research Award, and a grant from FuriosaAI.

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

# Appendix

## A   Effectiveness of our method in improving text-to-image models' counting fidelity

We provide more examples to show the effectiveness of applying our method to Stable Diffusion (Rombach et al., 2021) to see if it can improve the counting fidelity of the text-to-image generation model. We show results from 3 prompts, where for each prompt, 30 images are generated with 30 unique random seeds. To compare our method with the unmodified Stable Diffusion baseline, images in the same row are generated using the same random seed. It is worth noting that our method is not always effective. However, it does increase the likelihood of Stable Diffusion generating images with the correct object count.

| "**three** lions" | | "vintage silver plate tablespoons, serving spoon set of **two**" | | "An old building with ruined walls and **four** antique pink armchairs" | |
|---|---|---|---|---|---|
| Original | Embedding edited | Original | Embedding edited | Original | Embedding edited |

| "**three** lions" | | "vintage silver plate tablespoons, serving spoon set of **two**" | | "An old building with ruined walls and **four** antique pink armchairs" | |
|---|---|---|---|---|---|
| Original | Embedding edited | Original | Embedding edited | Original | Embedding edited |

| "**three** lions" | | "vintage silver plate tablespoons, serving spoon set of **two**" | | "An old building with ruined walls and **four** antique pink armchairs" | |
|---|---|---|---|---|---|
| Original | Embedding edited | Original | Embedding edited | Original | Embedding edited |

