# OpenReview forum: "Improving CLIP Counting Accuracy via Parameter-Efficient Fine-Tuning"
_TMLR — Accepted by TMLR_

### Review · Reviewer_q5MW · 2024-10-08

**Summary Of Contributions:**

The paper introduces a novel (and very parameter-efficient) method for improving pre-trained CLIP's counting accuracy, experimentally explores different text embedding editing techniques, introduces two new datasets for training and evaluating CLIP's counting accuracy, and shows how the presented approach can be combined with Stable Diffusion for generating images with accurate counts of objects. The proposed fine-tuning method only trains nine counting vectors. This is far fewer parameters than prior approaches to object counting. The paper focuses on counting at most 10 objects, and only evaluates the image generation approach qualitatively.

**Audience:**

Yes

**Broader Impact Concerns:**

No ethical concerns.

**Claims And Evidence:**

No

**Requested Changes:**

1. Add a discussion in the related work citing and discussing prior open-world counting approaches including the ones mentioned in the "Strengths and Weaknesses" section. Please emphasize how your approach is better than prior approaches to fine-tuning a low number
of parameters for counting. For example, CLIP-Count uses visual prompt tuning to fine-tune only a subset of parameters for the counting task. Why is your approach better than this? CLIP-Count does not limit the number of objects it can count. Please also discuss why your datasets are necessary when much larger and more diverse counting datasets containing images with hundreds of objects (e.g., FSC-147) exist. **[CRITICAL]**
2. Provide a quantitative evaluation of the Stable Diffusion approach for generating images with exact counts. For example, provide a set of text descriptions (maybe the ones from CountBench) and evaluate the number of objects generated by your approach versus the number of objects generated by the baseline and provide the resulting mean absolute errors for each approach. Since prior open-world counting methods already exist, I think the image generation is the most interesting and useful part of the paper, so it is important to evaluate it thoroughly. **[CRITICAL]**
3. Please discuss why CLIP is more accurate at counting certain categories of objects than others. I see that you do mention that images of cats and dogs are very common. I would like you to take this further and discuss this topic in the context of 'open-world.' How well does CLIP perform on categories outside of its large pretraining dataset. **[WOULD STRENGTHEN THE WORK]**
4. Please correct the following grammatical errors: **[CRITICAL]**
a) bottom of page 4, last paragraph: 'Similarity' should be 'similarly'
b) Intro to section 3 includes grammatical error:
"In Section 3.6, we will introduce how we collect and apply new counting datasets. Prior to the main method sections, we will first give a brief **[missing word]** of CLIP models in Section 3.1 and define the counting problem and how to evaluate CLIP’s counting ability in following Section 3.2."

**Strengths And Weaknesses:**

Strengths:
- The paper is well-written and easy to follow. It only has a few grammar mistakes (which I will specify
in the requested changes).
- The proposed method is parameter-efficient compared to the full fine-tuning of CLIP.
- The proposed method can be used to generate accurate counts of objects with Stable Diffusion. This
could be useful for generating synthetic data for object counting.
- Focuses on CLIP, allowing the paper to provide a deep dive rather than a superficial investigation.
- The experimental evaluation of counting 2-10 objects is thorough, covering 3 different CLIP backbones and reporting
results averaged over three random seeds.

Weaknesses:
- The paper does not consider counting more than 10 objects.
- The related work does not provide a sufficient overview of prior open-world counting methods. These include CLIP-Count [1], CounTX [2], VLCounter [3] and other related works that use CLIP for object counting. Furthermore, unlike the proposed approach, these prior techniques can count any number of objects. For example, the test set these approaches are evaluated on includes images with over 1000 objects. Therefore, the paper needs to make it clearer why the proposed approach is useful over these prior methods.
- The datasets that are proposed contain at most 10 objects and are limited in the categories of objects they contain. The paper does not mention FSC-147 [4] which is a standard counting benchmark that has images with thousands of objects and over 100 different categories of objects.
- The paper does not evaluate the Stable Diffusion image generation method quantitatively.
- The paper lacks a discussion on why certain categories are more difficult for CLIP to count. For example, perhaps the categories with the highest counting accuracy are in CLIP's large pretraining dataset. It would be good to connect this disucussion to the concept of 'open-world.'
- A few grammar mistakes

[1] R. Jiang, L. Liu, and C. Chen. Clip-count: Towards text-guided zero-shot object counting. In Proceedings of the ACM Multimedia Conference, 2023.

[2] N. Amini-Naieni, K. Amini-Naieni, T. Han, and A. Zisserman. Open-world text-specified object counting. In Proceedings of the British Machine Vision Conference. British Machine Vision Association, 2023.

[3] S. Kang, W. Moon, E. Kim, and J.-P. Heo. Vlcounter: Text-aware visual representation for zero-shot object counting. In Proceedings of the AAAI Conference on Artificial Intelligence, 2024.

[4] V. Ranjan, U. Sharma, T. Nguyen, and M. Hoai. Learning to count everything. In Proceedings of the IEEE Conference on Computer Vision and Pattern Recognition, 2021.

---

> ### Author Response · Authors · 2024-11-13
> **Responses to Reviewer q5MW**
>
> **Add a discussion in the related work citing and discussing prior open-world counting approaches including the ones mentioned in the "Strengths and Weaknesses" section. ... [CRITICAL]**
>
> => Response: We appreciate the reviewer’s valuable insights and suggestions. Our approach to counting differs fundamentally from prior work such as CLIP-Count, particularly in our task formulation. Unlike density estimation methods evaluated with mean square error (MSE), our method treats counting as a classification task. When we applied CLIP-Count to small-count scenarios—such as those in our dataset—it exhibited substantial errors, as density estimation does not align well with requirements for precise, low-count accuracy. The classification approach allows us to evaluate performance in terms of specific, discrete quantities, emphasizing the model's reasoning and capacity to identify distinct object instances.
> Our datasets, DiverseCount and ObjectCount, address CLIP’s limitations in object-specific and low-count contexts. While larger datasets such as FSC-147 cover images with high object counts, they align well with density estimation approaches and diverse contexts but do not offer the targeted low-count settings and object-specific evaluations that our datasets provide. These datasets were designed to enhance CLIP’s performance where exact small-count accuracy and object-type differentiation are crucial, providing insights into the model’s fundamental object recognition and reasoning abilities. We added a section in related works, "Density Estimation Framework vs. Classification-Based Approaches in Object Counting," and included CLIP-Count evaluation results on our dataset.
>
> **Provide a quantitative evaluation of the Stable Diffusion approach for generating images with exact counts... [CRITICAL]**
>
> => Response: We appreciate your suggestion to provide a quantitative evaluation of Stable Diffusion's capability in generating images with exact object counts. In accordance with your recommendation, we selected 10 prompts from CountBench and adjusted counting numbers within them as inputs for Stable Diffusion. We then quantitatively assessed images generated by our method versus those by the original CLIP model, evaluating if our approach more accurately generates images with correct counts. Results of this evaluation are presented in the revised paper under the section, "Effectiveness of our method in improving text-to-image models’ counting fidelity."
> In brief, we observed that our method significantly improves counting accuracy for prompts with counts from 2 to 5, generating images with correct object counts more reliably without additional training. However, for counts from 6 to 10, effectiveness diminished, likely due to the pretrained CLIP model’s limited text labels for higher numbers. More details are provided in the revised paper.
>
> **Please discuss why CLIP is more accurate at counting certain categories of objects than others. I see that you do mention that images of cats and dogs are very common. I would like you to take this further and discuss this topic in the context of 'open-world.' How well does CLIP perform on categories outside of its large pretraining dataset. [WOULD STRENGTHEN THE WORK]**
>
> => Response: We appreciate the reviewer’s feedback on CLIP’s varied counting performance across object categories. To address this, we expanded our discussion to analyze why CLIP counts certain objects more accurately, emphasizing the impact of pretraining data distribution. Common categories like cats and dogs are more accurately counted due to their frequent appearance in CLIP’s large-scale pretraining dataset, which enables better visual and semantic representations. Additionally, we discuss how data imbalance affects CLIP’s generalization to less common categories in open-world scenarios and note the potential need for targeted fine-tuning to address these gaps.
>
> **Please correct the following grammatical errors: [CRITICAL] a) bottom of page 4, last paragraph: 'Similarity' should be 'similarly' b) Intro to section 3 includes grammatical error: "In Section 3.6, we will introduce how we collect and apply new counting datasets. Prior to the main method sections, we will first give a brief [missing word] of CLIP models in Section 3.1 and define the counting problem and how to evaluate CLIP’s counting ability in following Section 3.2."**
>
> => Response: Thank you for pointing it out. We have corrected these grammar errors as suggested.

---

> > ### Comment · Reviewer_q5MW · 2024-11-25
> > **Feedback on Response**
> >
> > Dear Authors, thank you for taking the respond to each of my comments. I think that the responses strengthen the paper significantly, especially the stronger focus on the low-count setting (together with the CLIP-Count results) and the quantitative comparison for image generation. I also really appreciate the comparison to density map-based approaches.

---

> > > ### Author Response · Authors · 2024-12-07
> > > **Responses to Reviewer q5MW**
> > >
> > > We would like to once again express our sincere gratitude to the reviewer for the valuable suggestions provided. They have greatly helped us in further refining our work, and we are pleased to know that you find the improvements satisfactory.

---

### Review · Reviewer_RVAC · 2024-10-18

**Summary Of Contributions:**

This paper proposes learning-based and zero-shot methods to improve CLIP counting accuracy while retaining other non-counting related pre-trained knowledge and develops two object-counting benchmarks. Quantitative experiments on these two benchmarks and qualitative experiments employing the Stable Diffusion show that the proposed method improves CLIP's object counting accuracy across diverse model scales.

**Audience:**

Yes

**Claims And Evidence:**

No

**Requested Changes:**

see the weakness

**Strengths And Weaknesses:**

# Strength
1. This paper improves CLIP counting accuracy via training a few parameters.
2. Visualizations show that editing text embeddings is an effective way to enhance counting knowledge.

# Weakness
1. This method introduces nine counting vectors to manipulate the CLIP's text embedding. Each class corresponds to a certain number. There is a question of whether the method would still be practical with the number of objects increasing (e.g., 1000 objects in an image).

2. The paper proposes prompts-related counting vectors. It is related to the prompt-tuning parameter-efficient fine-tuning methods. More discussions are needed between the proposed method and other prompt-tuning methods[1,2,3].
[1] Visual prompt tuning.
[2] The power of scale for parameter-efficient prompt tuning.
[3] Prefix-Tuning: Optimizing Continuous Prompts for Generation.

3. It has been widely demonstrated that parameter-efficient fine-tuning paradigms can improve model performance with a few tunable parameters. This method follows the widely used techniques, which may hurt the novelty.

4. The novelty is limited. Previous methods [1,2] have provided explorations on improving CLIP's counting performance. For example, CrowdCLIP also transfers the counting problem into a classification problem by using CLIP. There needs to be more systemic comparisons with them.
[1] CrowdCLIP: Unsupervised Crowd Counting via Vision-Language Model.
[2] Teaching CLIP to Count to Ten.

5. The experiments are not convincing. This paper claims that they aim to improve the counting accuracy. However, they do not conduct experiments on the counting datasets. In other words, to evaluate the counting performance, the authors could conduct comparison experiments on typical counting benchmarks, such as the ShanghaiTech and UCF-QNRF.

---

> ### Author Response · Authors · 2024-11-13
> **Responses to Reviewer RVAC**
>
> **There is a question of whether the method would still be practical with the number of objects increasing (e.g., 1000 objects in an image).**
>
> => Response: We appreciate the reviewer’s thoughtful comment on the scalability of our method to higher counts. We acknowledge that our current approach is formed as a classification task that’s primarily designed for low-count scenarios, and directly extending it to scenarios with extremely high counts, such as 1000 objects, would be challenging. In response to this concern, we have expanded the discussion in the revised paper to address this limitation and to propose potential directions for future work.
>
> **The paper proposes prompts-related counting vectors. It is related to the prompt-tuning parameter-efficient fine-tuning methods. More discussions are needed between the proposed method and other prompt-tuning methods...**
>
> => Response: We appreciate the reviewer’s suggestion to expand our experiment section by comparing our approach to the prefix-tuning method. We didn’t include the comparison with Visual prompt tuning as in current paper, we focused on the potential on text encoder and how much performance can be gained by only modifying CLIP’s text encoder (so as well as Stable Diffusion’s text encoder). We found that prefix-tuning is harmful in current counting task. For more details, please checkout our update Experiment Section.
>
> **It has been widely demonstrated that parameter-efficient fine-tuning paradigms can improve model performance with a few tunable parameters. This method follows the widely used techniques, which may hurt the novelty.**
>
> => Response: We appreciate the reviewer's comment on the potential novelty concerns related to the use of parameter-efficient fine-tuning techniques. While it is true that parameter-efficient paradigms have been widely adopted to enhance model performance with minimal tunable parameters, our work contributes a novel application of this paradigm specifically tailored for improving counting tasks within CLIP models. What sets our method apart is how we adapt parameter-efficient fine-tuning to modify only the text encoder by training counting-specific vectors. Unlike conventional methods that often focus on broader adaptations or apply to tasks that do not require precise numerical understanding, our approach formulates counting as a classification problem and optimizes vectors that capture numerical semantics within the text embedding space. This targeted modification leverages the strengths of parameter-efficient fine-tuning while addressing the unique challenge of counting accuracy in vision-language models. Furthermore, we introduce zero-shot text embedding editing techniques that extend beyond typical fine-tuning, broadening the scope of parameter efficiency in enhancing performance without compromising the model's overall capabilities. To reinforce this distinction, we have expanded our discussion to emphasize the specific aspects of our approach that differ from existing parameter-efficient fine-tuning techniques, showcasing how our method focuses on numerical understanding and its impact on counting performance.
>
> **The novelty is limited. Previous methods [1,2] have provided explorations on improving CLIP's counting performance...**
>
> => Response: We thank the reviewer for pointing out relevant prior work on improving CLIP’s counting performance, such as CrowdCLIP. In response, we have added CrowdCLIP and other related approaches to the Related Work section and have clarified where our approach differs.
>
> **However, they do not conduct experiments on the counting datasets. In other words, to evaluate the counting performance, the authors could conduct comparison experiments on typical counting benchmarks, such as the ShanghaiTech and UCF-QNRF.**
>
> => Response: We appreciate the reviewer’s feedback regarding the use of standard counting benchmarks. We acknowledge that datasets like ShanghaiTech and UCF-QNRF are often used to evaluate high-density counting methods under metrics such as MSE. However, our work focuses on improving counting accuracy specifically in low-count, small-object scenarios, where we take counting as a classification task. Our datasets—DiverseCount and ObjectCount—were purpose-built to evaluate CLIP’s performance in low-count contexts, highlighting challenges in discrete, object-specific counting where tasks are treated as classification problems rather than density estimation. While benchmarks like ShanghaiTech and UCF-QNRF are valuable for density estimation in high-object-count scenarios, they are not directly aligned with our focus on precise low-count accuracy and object differentiation. To address the reviewer’s concern, we have clarified in the revised paper why we selected DiverseCount and ObjectCount and how they align with our problem formulation.

---

> > ### Comment · Reviewer_RVAC · 2024-12-03
> >
> > Although the rebuttal is not totally convincing, I believe this paper is still valuable.
> >
> > I hope the authors add a limitation part to discuss why can not conduct experiments on the typical counting benchmarks.

---

> > > ### Author Response · Authors · 2024-12-12
> > > **limitation of not conducting experiments on the typical counting benchmarks**
> > >
> > > We appreciate the reviewer’s suggestion. While our current focus is on low-count, object-specific scenarios, we acknowledge the importance of these benchmarks and that they can also be adapted for classification tasks. We have added a discussion in the paper to highlight this limitation and suggest future work to explore applying our methods to these datasets to broaden the evaluation scope.

---

### Review · Reviewer_PeLj · 2024-10-30

**Summary Of Contributions:**

The paper focuses on improving the counting limitations of contrastively trained vision-language models like CLIP. The authors propose and study methods (both fine-tuning and zero-shot) for improving counting by learning linear direction in text embedding space that correspond to specific counts. The authors show that these methods outperform full model fine-tuning in the small data regime while maintaining CLIP's broad capabilities.

**Audience:**

Yes

**Broader Impact Concerns:**

No concerns

**Claims And Evidence:**

Yes

**Requested Changes:**

Where feasible, please address the non-minor points listed under weaknesses.

**Strengths And Weaknesses:**

Strengths
 - The proposed methods are simple and usable even in a small-scale data regime, or even without data (zero-shot text embedding editing).
 - The authors introduce two novel datasets for fine-tuning and evaluating counting accuracy
 - The paper also studies downstream use-cases of edited CLIP text encoders such as in Stable Diffusion
 - In general, the paper is easy to read. Provided figures 1-4 are helpful in terms of understanding the main contributions of the paper

Weaknesses
 - Studied counts are limited to the range 2 to 10. In general, the methods do not generalize beyond the counts that have been explicitly trained.
 - The method is limited to a single counted quantity. A query like "five dogs and three cats" would not be directly feasible; at least it is not discussed how.
 - The paper focuses exclusively on the text encoder and implicitly assumes that the image encoder has the ability for counting (since it is not adapted). While the authors demonstrate that a pure focus on the text encoder can actually improve counting performance, a more detailed discussion and investigation of the image encoder's ability/limitations in terms of counting would be helpful.
 - The proposed datasets are not made publicly available. Also code for the methods is not provided
 - The studied CLIP models are relatively small and not state-of-the-art. To which extent would the gains observed by the authors transfer to stronger/more recent paper that are intrinsically stronger in counting?
 - Presentation of results in Section 4 and 5 could be improved; for instance: Table 2 does not highlight the best setting in bold, Figure 5 could use a range for the y-axis that focuses on the relevant part rather than showing 0-100% range,  Table 8 mixes non-comparable results on two different datasets into one table
 - Evaluation on Stable Diffusion is anecdotal, only evaluating the method on five (cherry-picked?) prompts.
 - As also discussed by the authors in Section 6, while the proposed methods are shown to be effective in the experiments, a better understanding of why and when this is expected to be the case remains open.
 - minor: Section 2 makes it sound as if OpenAI CLIP was trained on LAION-400M
 - minor: the studied diffusion model "Stable Diffusion 1.4" is somewhat outdated as of now.
 - minor: formatting of Section 3.2 is ugly (heading at the bottom of the page etc.). Also positioning of tables in the middle of the page in Section 4 is confusing.

---

> ### Author Response · Authors · 2024-11-13
> **Responses to Reviewer PeLj**
>
> **Studied counts are limited to the range 2 to 10. In general, the methods do not generalize beyond the counts that have been explicitly trained**
>
> => Response: We appreciate the reviewer’s comment regarding the limited range of counts (2 to 10) in our work and the generalization of our methods beyond explicitly trained counts. This range was selected due to practical dataset constraints and the challenges of collecting high-quality images with precise larger counts, ensuring reliable evaluations within a controlled scope. We have added a paragraph to the discussion acknowledging this limitation and noting that while our methods enhance counting accuracy within the trained range, they may not generalize seamlessly to higher counts. Future work could extend our approach to larger counts and investigate techniques like hierarchical counting representations or synthetic data to train on a wider range of counts.
>
>
> **The method is limited to a single counted quantity. A query like "five dogs and three cats" would not be directly feasible; at least it is not discussed how.**
>
> => Response: We acknowledge that our approach does not directly support queries with multiple counted quantities, such as “five dogs and three cats.” We added a discussion in the paper recognizing this limitation and suggesting potential directions, such as compositional embedding strategies or relational modeling, to extend our method for handling multi-object counting tasks.
>
>
> **The paper focuses exclusively on the text encoder and implicitly assumes that the image encoder has the ability for counting (since it is not adapted). While the authors demonstrate that a pure focus on the text encoder can actually improve counting performance, a more detailed discussion and investigation of the image encoder's ability/limitations in terms of counting would be helpful.**
>
> => Response: We appreciate the reviewer’s thoughtful comment on our exclusive focus on the text encoder and the implicit assumption of the image encoder's counting ability. Our approach of modifying only the text encoder is based on the hypothesis that CLIP’s counting limitations may stem from an inadequate representation of numerical concepts in the text embedding space. By addressing this, we demonstrate that counting improvements can be achieved without altering the image encoder. Additionally, focusing on the text encoder allows compatibility with Stable Diffusion models. While our results show that enhancing the text encoder significantly boosts counting performance, we do not exclude the possibility of image encoder limitations in certain cases. We have added a brief discussion acknowledging the potential influence of the image encoder on counting performance, suggesting future work could explore combined approaches that adapt both encoders for enhanced counting capabilities.
>
>
> **The proposed datasets are not made publicly available. Also code for the methods is not provided**
>
> => Responses: for the purpose of double-blinded review, we choose to release the code and dataset after the paper is published. We put a link to anonymous repo as temporary reference: https://anonymous.4open.science/r/clip_count_small_count-C7F6
>
>
> **The studied CLIP models are relatively small and not state-of-the-art. To which extent would the gains observed by the authors transfer to stronger/more recent paper that are intrinsically stronger in counting?**
>
> => We have added a more advanced model including SigLIP and BLIP in the ablation section and test the performance of zero-shot methods in our paper. We observed that some of the zero-shot methods indeed bring performance gain on these models. Please refer to Ablation section 5.3 for more details.
>
>
> **Presentation of results in Section 4 and 5 could be improved**
>
> => Response: We have improved the presentation by highlighting the best setting in bold in Table 2 (now Table 3); narrowing the y-axis range in Figure 5 to 10-70% to focus on relevant values, and splitting Table 8 (now Table 9) into two subtables by dataset.

---

> ### Author Response · Authors · 2024-11-13
> **Responses to Reviewer PeLj (continued)**
>
> **Evaluation on Stable Diffusion is anecdotal, only evaluating the method on five (cherry-picked?) prompts.**
>
> => Response: Thank you for your feedback regarding the evaluation of our method on Stable Diffusion. In response, we have conducted a more comprehensive quantitative evaluation. We selected 10 prompts from CountBench and adjusted counting numbers within them as inputs for Stable Diffusion. We then quantitatively assessed images generated by our method versus those by the original CLIP model, evaluating if our approach more accurately generates images with correct counts. Results of this evaluation are presented in the revised paper under the section, "Effectiveness of our method in improving text-to-image models’ counting fidelity." In brief, we observed that our method significantly improves counting accuracy for prompts with counts from 2 to 5, generating images with correct object counts more reliably without additional training. However, for counts from 6 to 10, effectiveness diminished, likely due to the pretrained CLIP model’s limited text labels for higher numbers. More details are provided in the revised paper on pages 11 and 12.
>
> **As also discussed by the authors in Section 6, while the proposed methods are shown to be effective in the experiments, a better understanding of why and when this is expected to be the case remains open.**
>
> => Response: We recognize that while our results demonstrate significant improvements in counting accuracy, especially in small-count tasks, understanding why and under what conditions these improvements occur requires further investigation. We expanded our discussion in Section 6, suggesting that targeted modifications in the text embedding space likely capture numerical semantics better for counting. However, understanding the exact mechanisms behind this improvement is an open question. Future work could include ablation studies on text and image encoder components, examining model behavior across object types and visual contexts, and developing theoretical frameworks to clarify observed performance gains.
>
> **minor: Section 2 makes it sound as if OpenAI CLIP was trained on LAION-400M**
>
> => We intended to clarify that one baseline, “Teaching CLIP to Count to Ten,” uses a dataset derived from LAION-400M.
>
> **minor: formatting of Section 3.2 is ugly (heading at the bottom of the page etc.). Also positioning of tables in the middle of the page in Section 4 is confusing.**
>
> => changed

---

> > ### Comment · Reviewer_PeLj · 2024-11-20
> > **Feedback on response**
> >
> > I would like to thank the authors for taking my review into account. The newly added content strengthens the paper. I would encourage the authors to proof-read the new content once more - there are issues like missing punctuation, whitespace, wrong citation style (cited vs. citep)  or with capitalization.
> > Lastly, I think the authors should tone down the claims regarding the new Table 11 - the improvement vs. SigLIP original is not clear as there is no setting which consistently outperforms the baseline

---

> ### Author Response · Authors · 2024-12-07
> **Responses to Reviewer PeLj**
>
> We sincerely appreciate the reviewer’s continued comment, which has helped us further refine our work. Following your latest suggestions, we have adjusted our claims regarding the results in Table 11 to acknowledge that our approach does not consistently improve the SigLIP baseline in all settings. This revision can be found on page 14 of our updated manuscript. Additionally, we have carefully proofread and corrected various formatting, punctuation, and citation style issues, and we have uploaded the revised version accordingly. Thank you again for your valuable feedback!

---

### Decision · Action_Editor_2Duj · 2024-12-10

**Recommendation:** Accept with minor revision

**Comment:**

Initially, reviewers appreciated the simplicity of the proposed approaches (PeLj, q5MW) and benchmarks (PeLj), the achieved results (RVAC), the downstream use cases (PeLj, q5MW), and the presentation (PeLj, q5MW). However, they also raised concerns, especially on the limited number of objects the models can handle (PeLj, RVAC, q5MW), missing experiments on standard counting datasets (RVAC, q5MW), the used CLIP models (PeLj), and missing discussions on the method choices (PeLj, RVAC) and challenges (q5MW).

The response and the revision addressed the main concerns by acknowledging the limitations and changing the focus on low-count settings, while also providing additional experiments and discussion of related works. All reviewers found the revision to be appropriate for this venue, and the AE agrees with their assessment.

As a final, minor modification, it would be helpful to clearly elaborate (e.g., the limitations or Sec. 4.1) on the choice of datasets (and why others have not been adopted, e.g., due to the focus etc.).

**Audience:**

Due to the widespread use of vision-language models and the necessity to address their shortcomings, the article will be of interest to multiple researchers in the field of computer vision and multimodal learning.

**Claims And Evidence:**

The article tackles the problem of counting in CLIP, and designing text embedding editing techniques for it. These techniques are either zero-shot or based on fine-tuning. The article also proposes a dataset for training and evaluating counting in CLIP. both zero-shot and fine-tuning-based solutions for it.

The main claim of the article is that the proposed strategies can improve CLIP counting accuracy without harming its other zero-shot capabilities. While the effectiveness is limited to scenarios with a low number of objects, the claim is supported via multiple experiments on two counting datasets and standard zero-shot classification benchmarks.